# How Control Information Influences Multilingual Text Image Generation and Editing?

**Boqiang Zhang, Zuan Gao, Yadong Qu, Hongtao Xie**[*]
University of Science and Technology of China
{cyril,zuangao,qqqyd}@mail.ustc.edu.cn   htxie@ustc.edu.cn

## Abstract

Visual text generation has significantly advanced through diffusion models aimed at producing images with readable and realistic text. Recent works primarily use a ControlNet-based framework, employing standard font text images to control diffusion models. Recognizing the critical role of control information in generating high-quality text, we investigate its influence from three perspectives: input encoding, role at different stages, and output features. Our findings reveal that: 1) Input control information has unique characteristics compared to conventional inputs like Canny edges and depth maps. 2) Control information plays distinct roles at different stages of the denoising process. 3) Output control features significantly differ from the base and skip features of the U-Net decoder in the frequency domain. Based on these insights, we propose TextGen, a novel framework designed to enhance generation quality by optimizing control information. We improve input and output features using Fourier analysis to emphasize relevant information and reduce noise. Additionally, we employ a two-stage generation framework to align the different roles of control information at different stages. Furthermore, we introduce an effective and lightweight dataset for training. Our method achieves state-of-the-art performance in both Chinese and English text generation. The code and dataset are available at https://github.com/CyrilSterling/TextGen.

## 1 Introduction

With the development of diffusion-based generative models [9, 26, 20] and image-text paired datasets [23, 8], significant improvements have been made in the quality of image generation. Given the prevalence of text in natural scenes (e.g., posters, slides, signs, book covers, etc.), generating images containing text accurately and reasonably is crucial.

Recently, several methods have been proposed to address the generation of high-quality visual text images [33, 14, 4, 28]. Among these, ControlNet-based approaches show strong potential [33, 28], enabling flexible multilingual visual text generation, text position control, and easy integration into existing pre-trained diffusion models. Current methods directly utilize ControlNet [36] for text generation control, using a global glyph image of a standard font as the condition (as shown in Figure 1). However, achieving accurate and robust control remains challenging due to the complex and fine-grained structure of characters. Hence, we pose a meaningful question: *How does control information influence multilingual text image generation?*

To address the above issue, we investigate the impact of control information on visual text generation from three perspectives, as illustrated in Figure 1. **For the input of control information**, the current model uses a glyph image to guide the generation of accurate text textures. However, unlike general ControlNet inputs such as Canny edges, depth, or M-LSD lines, text glyph images have unique

---

[*]Corresponding author

38th Conference on Neural Information Processing Systems (NeurIPS 2024).

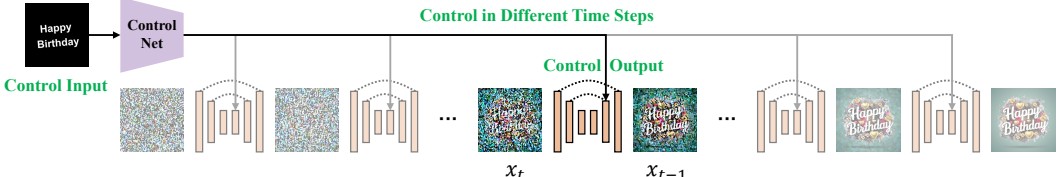

Figure 1: The overall pipeline of recent text generation works. It utilizes a ControlNet for guiding the text generation process, employing a glyph image with a standard font as the control information. Control information at different stages is generated in the same manner and directly added to the skip features in the U-Net decoder.

properties: 1) Glyph images have areas of high information density concentrated within specific regions, with the rest being meaningless backgrounds. 2) Generation in text region is fine-grained, but extracting fine-grained features from glyph images using standard convolution methods is challenging (further discussed in Sec. 3.1.1). These properties limit the performance. **For the control information in different time steps**, current models follow ControlNet [36] by using fixed control, but they often overlook the role of different time steps in the generation process. We further explore the impact of control at various steps in Sec. 3.1.2. Control in early steps influences both text and non-text regions, ensuring that the background reasonably matches the text. Control in late steps still plays a significant role in modifying mistakes, which is different from the general generation [2, 15]. **For the output of control feature**, these features are injected into the U-Net decoder, which receives three types of features: base, skip, and control. Each of these components differs in the frequency domain, which explains their respective roles (discussed in Sec. 3.1.3). Balancing these components during inference is crucial. Overall, we explore the influence of control information in text generation, raising several critical questions essential for advancements in visual text generation.

Based on the analysis above, we introduce TextGen, a novel framework aimed at enhancing the quality of control information. Specifically, for control input, we introduce a Fourier Enhancement Convolution (FEC) block to extract spatial and frequency features from the glyph control image. This operation can enhance the perception of useful regions and edge textures. For the output control feature, we introduce a frequency balancing factor to adjust the frequency information among the features during inference. For the control information in different stages, we propose a two-stage framework for coarse-to-fine generation. This framework trains the first-stage model for global control and the second-stage model for detail control. Based on the two-stage framework, we naturally propose a novel inference paradigm for unifying text generation and editing tasks. Furthermore, as current datasets for visual text generation are large-scale and noisy, we construct a lightweight but high-quality dataset for effective training (details provided in Appendix A). Unlike previous works, we were the first to delve into control information in the visual text generation task. Our framework enhances the quality of detail generation while elegantly achieving unified generation and editing tasks. To summarize, our contributions are as follows:

- We conducted an in-depth study and discussion on the impact of control information in visual text generation tasks. Our findings can inspire more future research in this area.
- We propose a framework for multilingual visual text generation and editing based on our analysis, which contains a two-stage pipeline and a Fourier enhancement in both training and inference. This framework achieves state-of-the-art performance.
- We construct an open-source effective and lightweight dataset for the training of visual text generation and editing.

## 2 Related Works

### 2.1 Text Generation

With the advancement of denoising diffusion probabilistic models [9, 26] and text-to-image generation [22, 2, 20], it has become possible to generate high-quality images. However, visual text generation remains challenging due to the need for fine-grained alignment and character detail representation. Recent studies, such as Imagen [22] and eDiff-I [2], have focused on improving text

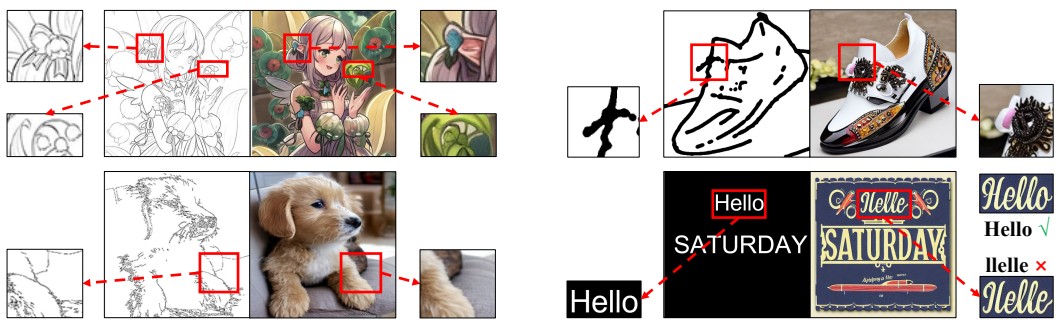

Figure 2: Differences between text control information and general ControlNet control information, including anime line drawings, M-LSD lines, and Canny edges. General controls focus on the overall structure and tolerate localized errors, while text control requires precise detail.

generation from the perspective of text encoders. They found that T5-based encoders [19] outperform CLIP text encoders [18]. GlyphDraw [14] presents a robust baseline for visual text generation by incorporating conditions. It utilizes a glyph image representing a single word as the content condition and a mask image as the positional condition. However, GlyphDraw is limited to generating only one text line per image. GlyphControl [33] further introduces a ControlNet-based framework that employs a global glyph image as the condition, providing both glyph and positional information, achieving outstanding generation performance. Glyph-ByT5 [13] fine-tunes a T5 language model for paragraph-level visual text generation, achieving remarkable performance in dense text generation. However, it is restricted to producing text in English. We propose an effective multilingual framework by controlling information enhancement in a ContorlNet-based framework.

## 2.2 Text Editing

Scene text editing aims to replace text in a scene image with new text while preserving the original background and style. Early approaches focused on generating text on cropped images, allowing for more precise text area generation [34, 10, 11, 24, 30, 17]. SRNet [30] was the first to divide the editing task into three sub-processes: background inpainting, text conversion, and fusion, which inspired subsequent works [17, 21, 32]. Although these methods achieved excellent generation performance, integrating the cropped text area into the original scene images proved challenging such as edge inconsistency. Recently, leveraging the diffusion model, some approaches have conducted generation on complete scene images directly, without decomposing the task into sub-processes. DiffUTE [3] proposes a concise framework for directly generating the edited global scene image using the diffusion process. However, solely focusing on the complete editing task limits the practicality and generalization of the model.

## 2.3 Joint Text Generation and Editing

Due to the similarity between visual text generation and editing tasks, developing a unified framework to jointly address these tasks is meaningful. TextDiffuser [5, 4] employs a mask to indicate areas requiring editing, ensuring multitasking uniformity. During the generation task, the mask remains empty, while during the editing task, it preserves areas that do not require editing. Additionally, TextDiffuser introduces a layout generator to design the distribution of text lines. Similarly, Any-Text [28] adopts a comparable approach to maintain the uniformity of two tasks and further enhances generation quality with a text embedding module and perceptual loss. Building on our explorations, we propose a two-stage model and design a novel inference paradigm to achieve multitasking unity.

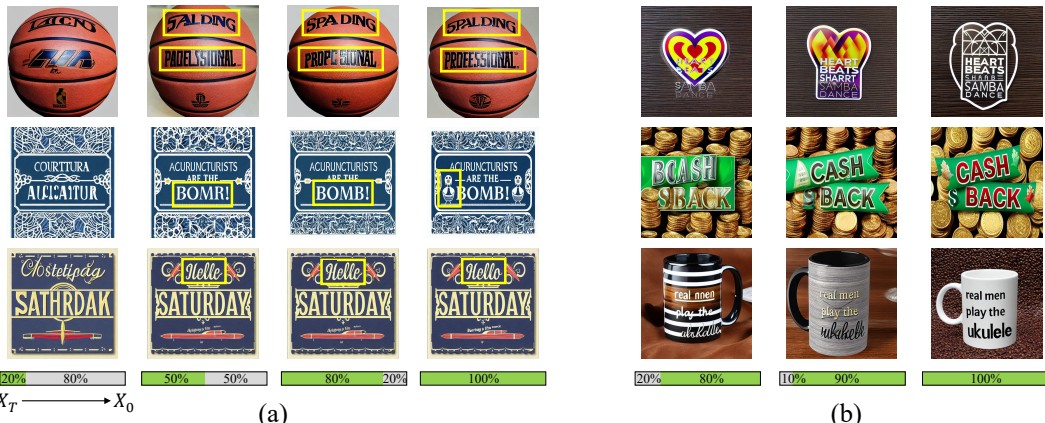

$X_T \longrightarrow X_0$       (a)       (b)

Figure 3: Visualization of the impact of control at different denoising stages. Control information is removed in the gray segments of the color bar during denoising. (a) Since visual text generation requires much detail texture, control information in later stages still plays an important role. (b) Even with only glyph and position images as control information, early-stage control influences non-text regions, ensuring the text region is coherent and matches the background.

## 3 Method

### 3.1 Motivations

#### 3.1.1 Control Information Input

In recent works, the inputs of the control module are the glyph image and position image, which provide the texture and position condition for text regions. However, different from general ControlNet conditions (e.g., Canny edge, M-LSD lines, depth, etc.), text conditions have distinct characteristics. As illustrated in Figure 2, general ControlNet conditions typically influence only macroscopic coarse-grained style and global edges, and some incorrect minor texture generation is considered acceptable. However, small differences in texture details can lead to content errors or unrealistic and unreasonable textures in visual text generation. Therefore, using the general ControlNet poses challenges in controlling detailed textures and fine-grained handwriting. Moreover, the text glyph condition concentrates only on certain regions, making it a sparse condition unlike other general conditions. This characteristic causes the convolution-based ControlNet encoder to introduce noise in empty areas when extracting features, which interferes with the text area's features and affects the correct allocation of attention between edge and background areas. Enhancing the ControlNet encoder's perception of locally useful details and edge information is essential for improving text image generation. Meanwhile, the characteristic of having high information density in local areas suggests that we can seek solutions in the frequency domain.

#### 3.1.2 Control Information in Different Denoising Stages

Some studies on general diffusion [15, 2] have suggested that control information in the later stages of the denoising process contributes little to the diffusion model. However, due to the specific nature of text, where text strokes constitute detailed information, we find that control in the later stages remains crucial. As shown in Figure 3 (a), omitting control information in the later stages often results in incorrect text content generation. The control module in these later stages corrects such errors, ultimately leading to high-quality images. This finding indicates that performing the editing task at a late time step is reasonable, as it mitigates the performance impact of joint multi-task training.

Furthermore, we investigate the role of early control. As shown in Figure 3 (b), the control information in early steps has a significant impact on the global semantics of the generated image, aiding in the alignment of text areas with the global scene. Without the control information in the early steps, the generated text regions appear unreasonable and do not match the background. Notably, even though only glyph and position images provide control information, the early stages still strongly influence the generation of non-text regions (the non-text areas may be totally different).

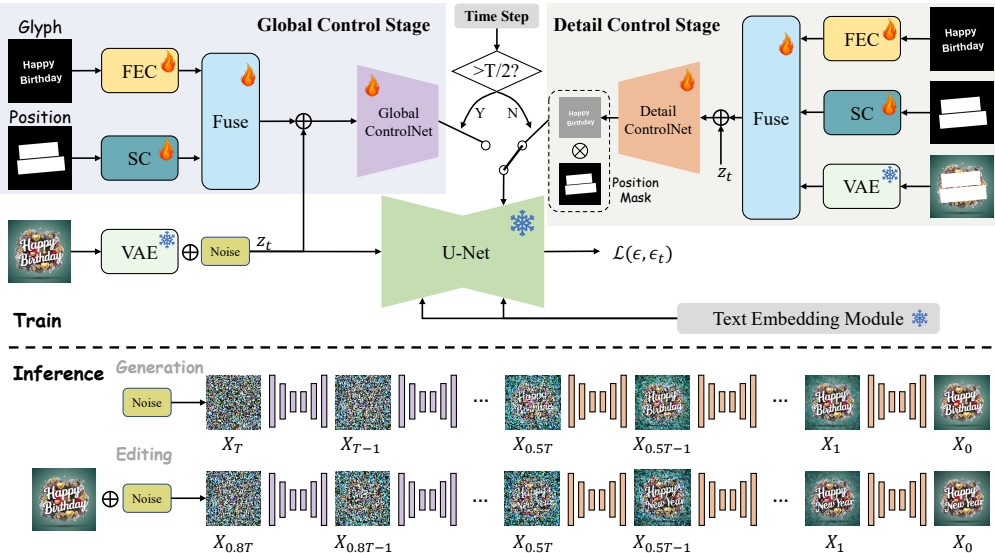

Figure 4: The pipeline of our TextGen. It comprises two stages: the global control stage and the detail control stage. Control information utilizes a Fourier Enhancement Convolution (FEC) block and a Spatial Convolution (SC) block to extract features. During inference, we introduce a novel denoising paradigm to unify generation and editing based on our framework design. Best shown in color.

Therefore, the control information of different stages should be adapted according to these findings. We divide the control module into global and detail stages (Fig. 4), each with distinct parameters. In the global control stage, we expect that the control information can affect the entire image, while in the detail control stage, we incorporate a mask to guide the model in optimizing local details. Furthermore, the detail stage can act as a refiner in text generation and as an editor in text editing.

### 3.1.3 Control Information Output

During inference, the output of the control module is injected into the base diffusion process. Each layer in the U-Net decoder can be formulated as: $\mathbf{F}_i = \mathcal{D}_i(\mathbf{F}_{i-1}, \mathbf{S}_{i-1}, \mathbf{C}_{i-1})$, where $\mathcal{D}_i$ is the $i$-th layer, $\mathbf{F}_i$ is the output feature of $i$-th layer, $\mathbf{S}_i$ and $\mathbf{C}_i$ represent the skip feature and the control feature of $i$-th layer, respectively. These three parts of the input represent different types of information. Following FreeU [25], we further investigate the balance among these three components. As shown in Fig. 5, we visualize the Fourier relative log amplitudes of $\mathbf{F}, \mathbf{S}$ and $\mathbf{C}$. It can be observed that the skip feature contains more high-frequency information than the base feature, which may infer denoising (the same conclusion with [25]). Therefore, there is a need for balancing between the base feature and the skip feature. Furthermore, compared with the fusion feature, the control feature has more high and low-frequency information, with a greater gap at low frequencies than at high frequencies. However, since we only aim to control the texture, which belongs to high-frequency information, the low-frequency information needs to be suppressed.

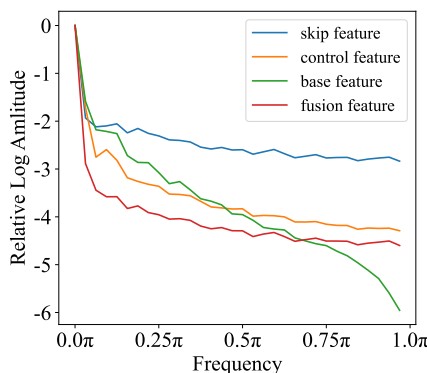

Figure 5: The relative log amplitude of three parts of features in U-Net decoder.

### 3.2 Pipeline

Based on the motivations and discussions above, we propose a novel framework named TextGen. The pipeline of our TextGen is illustrated in Figure 4. During training, our model comprises two stages: the global stage and the detail stage. The parameters of the pre-trained diffusion U-Net are

fixed, and each stage only trains a ControlNet. The global control stage is trained exclusively on larger time steps, while the detail control stage is trained only on smaller time steps. Through such an operation, the global control stage focuses on structure and style construction, and the detail control stage concentrates on detail modification. In this section, we first detail the control design in Sec. 3.3. Then, we describe the enhancement of control information output in Sec. 3.4. Finally, we propose a novel inference paradigm for task unification using our model in Sec. 3.5.

## 3.3 Model Control

**The global control stage** receives two pieces of control information: a position image indicating the text positions and a glyph image indicating the standard font of the texts. We use Spatial Convolution (SC) block and Fourier Enhancement Convolution (FEC) block to extract the feature of position image and glyph image, respectively. The structures of SC and FEC are illustrated in Figure 6. In SC, we employ general convolution for spatial perception, whereas in FEC, we use two branches for information extraction. The spatial branch is similar to SC, while the frequency branch employs a 2D Fast Fourier Transform (FFT) algorithm to transform the features into the frequency domain and performs convolution in this domain. Subsequently,

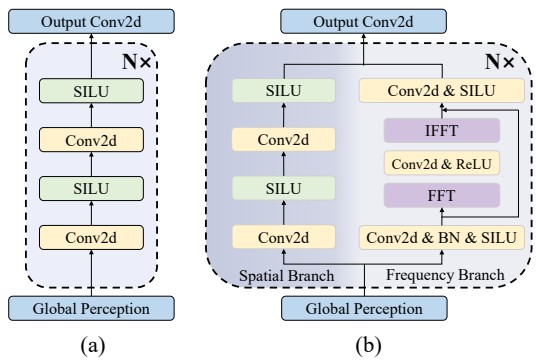

Figure 6: (a) The Spatial Convolution Block. (b) The Frequency Enhancement Convolution Block.

an Inverse Fast Fourier Transform (IFFT) algorithm is used to transform the frequency features back. Additionally, the input layer of both blocks is a global perception operation, achieved through convolution with a large kernel size. This is because the text contains global semantic information, and the general convolution-based encoder has a local receptive field that limits information interaction.

**The detail control stage** receives three pieces of control information: a position image, a glyph image, and a masked image. Unlike the global control stage, the detail control stage incorporates the masked image as one of its inputs, aiming to provide background information. This stage is designed to generate specified texts at designated positions while keeping the background consistent with the masked image. The module's output is multiplied by a position mask, enabling the model to focus on modifying the detailed texture of the text area. It's worth noting that the mask is only applied in the last two layers of the U-Net decoder (for feature sizes greater than or equal to 32). This is because the first two layers (feature sizes less than 32) primarily handle global information, which contributes little to the detailed texture. Moreover, by retaining the first two layers without applying the mask, we ensure that the background of the output image remains consistent with the masked image while modifying inconsistent areas.

**Discussion:** The frequency enhancement block performs convolution operations in both spatial and frequency domains, offering two main benefits: 1) The Fourier transform of visual features provides global information about the glyph image, overcoming the limitations of the receptive field in spatial convolution. 2) The glyph image contains both localized detail-rich areas and meaningless backgrounds. Convolution in the frequency domain acts as a frequency filter, allowing for the adjustment of attention to different frequency components. This facilitates the extraction of useful information and mitigates the interference of irrelevant information from a global perspective.

## 3.4 Enhancement for Control Information Output

In our framework, each layer of the U-Net decoder receives three parts of information: the backbone feature $\mathbf{F}$, the skip feature $\mathbf{S}$, and the control feature $\mathbf{C}$. As discussed in Sec. 3.1.3, there is a need for balancing among three parts during inference. Therefore, we propose a Fourier enhancement method as formulated as follows:

$$\mathbf{F}_{i+1} = \mathcal{D}_{i+1}([\mathbf{S}_i + \alpha \mathbf{C}'_i, \beta \mathbf{F}'_i]),$$
$$\mathbf{C}'_i = \mathscr{F}^{-1}(\mathscr{F}(\mathbf{C}'_i) \odot \gamma), \tag{1}$$

where $\mathscr{F}$ and $\mathscr{F}^{-1}$ represent FFT and IFFT, $\odot$ denotes the element-wise multiplication. $\alpha$ and $\beta$ are balancing factors for the control feature and the base feature. $\gamma$ is a modulation factor in the frequency domain. Although enhancing high-frequency information is desired, directly doing so will introduce noise and reduce generation quality. Therefore, we suppress low-frequency information to emphasize high-frequency components. This is achieved using a scalar $s$ to suppress low-frequency information as follows:

$$\gamma(r) = \begin{cases} s & \text{if } r < r_{thresh}, \\ 1 & \text{otherwise.} \end{cases} \tag{2}$$

### 3.5 Inference Paradigm for Multi-task

Based on our model and analysis, we propose a novel inference paradigm for task unification. **For the image generation task**, random noise is inputted into the global control stage for the early $T/2$ steps and the detail control stage for the remaining steps. **For the text editing task**, the original image is first noise-added to 80% time-step, which maintains most of the global style and texture while destroying the original text content. Then the new text content is first generated using the global control stage until the $T/2$ time step, and the remaining time steps use the detail control stage to modify the details. Since the control input of the detail control stage contains the masked original image, it can restore the background information that was destroyed during the noise addition, and at the same time optimize the new text content at the specified location.

## 4 Experiments

### 4.1 Datasets

Recently, several works have introduced datasets for text generation and editing tasks. TextDiffuser [5, 4] introduced a dataset named MARIO-10M, comprising approximately 10 million images annotated with bounding boxes and content of text regions. AnyText [28] proposed a benchmark named AnyWord for evaluation. However, training on 10 million images requires significant computing resources. Therefore, we introduce TG2M, a multilingual dataset sourced from publicly available images including MARIO-10M [5], Wukong [8], TextSeg [31], ArT [6], LSVT [27], MLT [16], ReCTS [37]. Although TG2M contains significantly fewer images, it is highly effective for training and achieves superior performance. The dataset will be detailed in the Appendix A.

### 4.2 Implementation Details

In our implementation, the diffusion model is initialized from SD1.5[2], and our code is based on diffusers[3]. The text embedding module follows AnyText [28]. We train our model on the TG2M dataset using 8 NVIDIA A40 GPUs with a batch size of 176. Our model converges rapidly and requires only 5 epochs of training. The learning rate is set to 1e-5. Following previous generation and recognition works [35, 29, 28, 7], we set the maximum length of each text line to 20 characters and the maximum number of lines in each image to 5. During inference, the Fourier balance factors $\alpha$, $\beta$, and $s$ are set to 1.4, 1.2, and 0.2, respectively.

We evaluate our model on the AnyWord [28] benchmark. We use DuGuangOCR [4] to recognize the text region and measure performance using sentence accuracy (ACC), Normalized Edit Distance (NED), and Fréchet Inception Distance (FID). During inference, the settings (random seed, control strength, etc.) are consistent across all experiments.

### 4.3 Ablation Study

Owing to resource constraints, following AnyText [28], we randomly select 200k images (40k English and 160k Chinese) from TG2M as the training set for ablation. The results are shown in Tab. 1.

---

[2]https://huggingface.co/runwayml/stable-diffusion-v1-5
[3]https://github.com/huggingface/diffusers
[4]https://modelscope.cn/models/iic/cv_convnextTiny_ocr-recognition-general_damo

Table 1: Ablation of proposed methods. FEC denotes Fourier enhancement convolution, GP signifies global perception in FEC, TS represents the two-stage generation framework, and IFE indicates inference Fourier enhancement.

| FEC | GP | TS | IFE | English | | Chinese | |
|---|---|---|---|---|---|---|---|
| | | | | ACC↑ | NED↑ | ACC↑ | NED↑ |
| | | | | 49.51 | 75.99 | 31.50 | 60.22 |
| ✓ | | | | 50.90 ↑1.39 | 76.81 ↑0.82 | 56.98 ↑25.48 | 77.28 ↑17.06 |
| ✓ | ✓ | | | 52.24 ↑1.34 | 77.64 ↑0.83 | 58.60 ↑1.62 | 78.04 ↑0.76 |
| ✓ | ✓ | ✓ | | 53.03 ↑0.79 | 78.14 ↑0.50 | 60.47 ↑1.87 | 78.86 ↑0.82 |
| ✓ | ✓ | ✓ | ✓ | **60.18** ↑7.15 | **82.28** ↑4.14 | **61.42** ↑0.95 | **80.56** ↑1.70 |

Table 2: Comparison with state-of-the-art methods. Data denotes the amount of data used in the training process. Our baseline is the AnyText-v1.0 [28] model trained on our TG-2M.

| Methods | Data | English | | | Chinese | | |
|---|---|---|---|---|---|---|---|
| | | ACC↑ | NED↑ | FID↓ | ACC↑ | NED↑ | FID↓ |
| ControlNet [36] | - | 58.37 | 80.15 | 45.41 | 36.20 | 62.27 | 41.86 |
| GlyphControl [33] | 10M | 52.62 | 75.29 | 43.10 | 4.54 | 10.17 | 49.51 |
| TextDiffuser [5] | 10M | 59.21 | 79.51 | 41.31 | 6.05 | 12.62 | 53.37 |
| AnyText-v1.0 [28] | 3.5M | 65.88 | 85.68 | **35.87** | 66.34 | 82.64 | **28.46** |
| Baseline | 2.5M | 64.26 ±0.51 | 84.80 ±0.10 | 41.65 ±2.84 | 65.02 ±0.11 | 81.95 ±0.12 | 30.04 ±0.70 |
| TextGen | 2.5M | **73.36** ±0.15 | **88.98** ±0.12 | 40.37 ±1.71 | **67.92** ±0.28 | **83.94** ±0.09 | 28.90 ±0.94 |

From the table, we observe the following: 1) All proposed methods yield performance gains in both Chinese and English text generation. 2) The FEC block enhances edge and texture features through Fourier enhancement, with more significant gains in Chinese due to the greater complexity of Chinese characters. It is worth noting that our model trained only on 200k images can achieve 61.42% and 60.18% sentence accuracy on Chinese and English text generation, which is almost as good as the state-of-the-art performance on large-scale training sets.

## 4.4 Comparison Results

### 4.4.1 Quantitative Results

Compared to other methods, our approach uses less data while outperforming the state-of-the-art, as shown in Table 2. For a fair comparison, all methods are evaluated under the same settings. The performance of both the baseline and TextGen is assessed using four random seeds, with the final metrics reported as averages and standard deviations. Our method achieves a 9.1% gain in sentence accuracy for English texts and a 2.9% gain for Chinese texts compared to our baseline trained on TG-2M. Notably, other approaches require large amounts of training data and employ perceptual loss to enhance performance, which is training-inefficient. Our method, in contrast, does not require additional losses and converges faster, making it easier to train. Besides, we compute the FID on the AnyWord-FID [28] benchmark. Our FID scores achieve comparable performance but not the best. The higher FID score does not necessarily imply the lower visual quality of the generated images. Our generated images demonstrate greater diversity and there is a distribution gap between our training set and AnyWord. This issue is discussed in more detail in Appendix C.

### 4.4.2 Qualitative Results

The qualitative comparison is shown in Fig.7. Our TextGen produces high-quality images with text in various scenarios and excels in generating artistic text with a wide range of visually appealing styles. For Chinese text, as demonstrated in Fig.8, TextGen generates more realistic and readable results, particularly in smaller texts. Finally, Fig. 9 illustrates the editing capabilities of our model, which can edit various text styles and contents using the proposed inference paradigm.

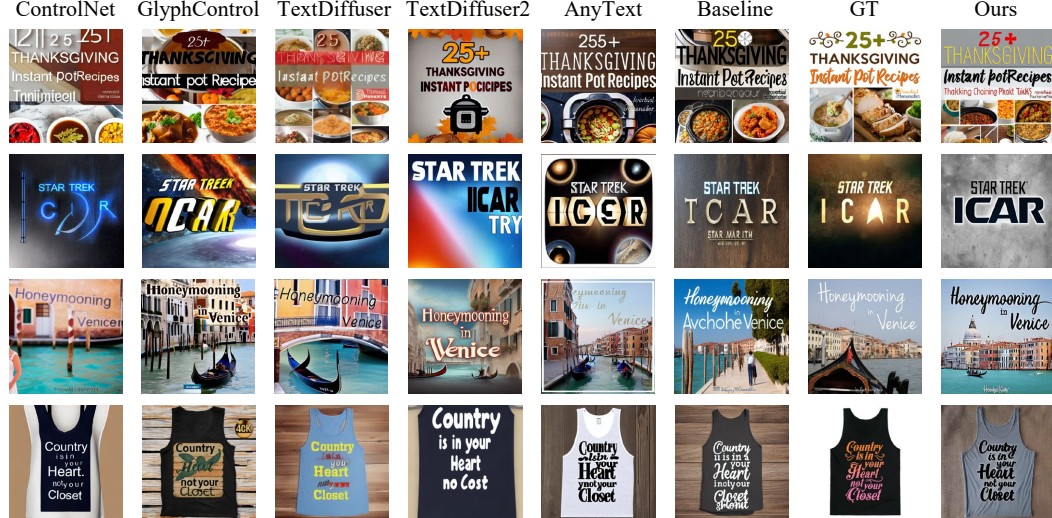

Figure 7: Qualitative comparison of generation performance in English texts. Our TextGen can generate more artistic and realistic texts.

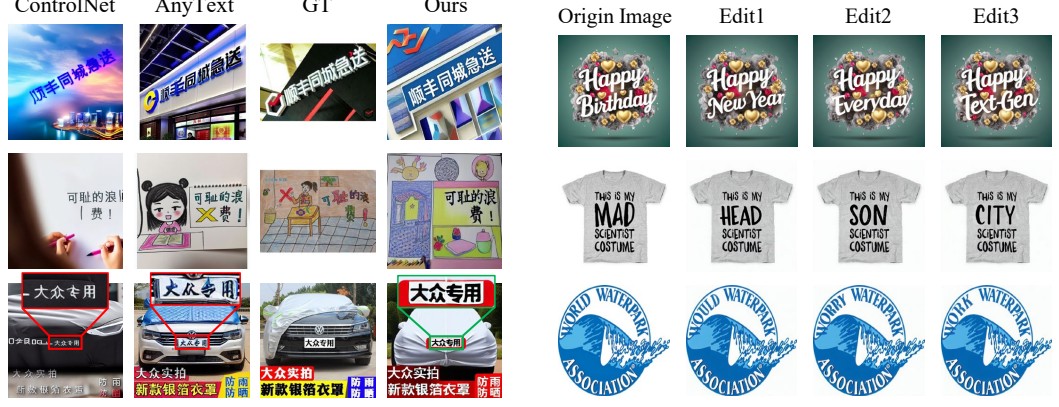

Figure 8: Comparison of generation in Chinese. Figure 9: Visualization of editing performance.

## 5 Limitations

We propose a novel diffusion-based network for multilingual text generation and editing, which demonstrates robust performance. Our network excels at generating high-quality scene images with text. However, the framework based on latent diffusion presents certain limitations. 1) The VAE employed in latent diffusion restricts the performance of fine-grained texture generation, particularly for complex texts. Because the diffusion process operates in the latent feature space, the VAE decoder struggles to generate small or complex texts. Consequently, our method is unable to generate such images. This issue can be addressed by generating each local sub-region separately. 2) The text condition controls the content of the generated image. However, the CLIP text encoder has limited ability to comprehend text, resulting in performance limitations. To resolve this issue, we can pre-train the diffusion model with a large language model serving as the text condition encoder.

Moreover, generating false text can contribute to the spread of misinformation, potentially resulting in serious consequences. It is hoped that this technology will be used responsibly, fostering a healthy and ethical academic environment.

# 6 Conclusion

Based on ControlNet, current visual text generation has made significant progress. In this work, we build on recent studies using ControlNet to investigate how control information influences visual text generation from three perspectives: control input, control at different stages, and control output. Through experiments and discussions, we derive several key conclusions. Based on our analysis, we propose a novel visual text generation framework that improves control information utilization, which surpasses the state-of-the-art performance. We believe the insights we gained about control information can inspire future research in the community.

# 7 Acknowledgments

This work is supported by the National Key Research and Development Program of China (2022YFB3104700), the National Nature Science Foundation of China (U23B2028, 62121002, 62102384). This research is supported by the Supercomputing Center of the USTC. We also acknowledge the GPU resource support offered by the MCC Lab of Information Science and Technology Institution, USTC.

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

# A  Details of Dataset

To train the visual text generation model, we constructed a dataset called TG-2M, which contains rich textual data. The images in this dataset are sourced from existing open-source datasets, including MARIO-10M [5], Wukong [8], ArT [6], LSVT [27], MLT [16], ReCTS [37], TextSeg [31]. Following AnyText [28], we filter the images with some rules, which can devided into three steps. Specifically, the filtering rules of the first step include:

- The images of width smaller than 256 will be filtered out.
- The images with aspect ratio larger than 2.0 or smaller than 0.5 will be filtered out.

After step 1, we use PP-OCR[5] to detect and recognize the texts in these images. Then, we undergo the second filtering step:

- The images with more than 10 texts will be filtered out.
- The images with more than 3 small texts will be filtered out. The small text refers to horizontal text with a height of less than 30 pixels or vertical text with a width of less than 30 pixels. The orientation of the text is determined by the aspect ratio of the text bounding box, with an aspect ratio less than 0.5 considered vertical text.
- Images containing more than 3 text instances with recognition scores below 0.7 will be filtered out.

We use BLIP-2 [12] and Qwen-VL [1] to recaption the images. First, we generate initial captions using BLIP-2. Because some initial captions are low-quality, we then modify these captions using Qwen-VL. The low-quality captions are defined as those containing meaningless text or having low CLIP similarity with the reference image. This process is necessary because some captions generated by BLIP-2 are meaningless, as shown in Fig. 10. Additionally, although Qwen-VL's captions are of high quality, many of them are quite lengthy, which can affect the understanding of the CLIP text encoder in the diffusion model.

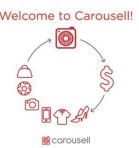

**BLIP-2 Caption:** carouselell - carouselell - carouselell - carouselell - carouselell - carouselell - carouselell - carouselell –

**Qwen-VL Caption:** An animated welcome message displayed by the website carousell.com. It features several icons arranged around a central circle which contains the text "Carousell".

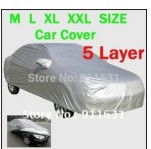

**BLIP-2 Caption:** car cover for car, car cover for car, car cover for car, car cover for car, car cover for car.

**Qwen-VL Caption:** Free shipping! Car Cover Sun UV Snow Dust Rain Resistant Protection Covers M L XL XXL size(China (Mainland))

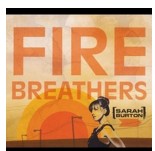

**BLIP-2 Caption:** barahan - fire breathers

**Qwen-VL Caption:** The album artwork of Sarah Burtons' debut release 'Firebreatherers'.

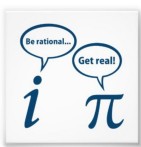

**BLIP-2 Caption:** i pi pi pi pi pi pi pi pi pi pi pi pi pi pi pi pi pi pi pi pi pi pi pi pi pi pi pi pi pi pi pi pi

**Qwen-VL Caption:** Two speech bubbles say be rational and get real.

Figure 10: Comparison of captions by BLIP-2 and Qwen-VL.

Examples from our TG-2M dataset are shown in Fig.11, illustrating the variety of image styles. The dataset statistics are summarized in Tab.3. TG-2M contains a total of 2.53 million images with 9.54

---

[5]https://github.com/PaddlePaddle/PaddleOCR

Table 3: The statistics of our proposed TG-2M dataset.

|          | image count | line count | mean chars/line | #line < 20 chars |
|----------|-------------|------------|-----------------|------------------|
| English  | 1.3M        | 5.59M      | 4.23            | 5.50M, 98.4%     |
| Chinese  | 1.23M       | 3.95M      | 5.68            | 3.83M, 97.0%     |
| Total    | 2.53M       | 9.54M      | 4.83            | 9.33M, 97.8%     |

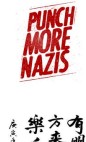 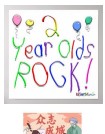 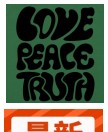 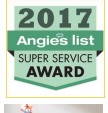 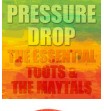 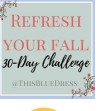 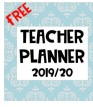 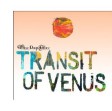 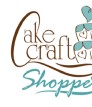
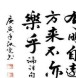 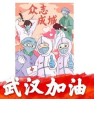 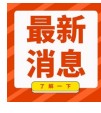 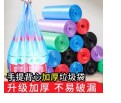 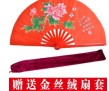 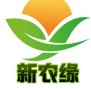 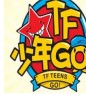 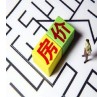 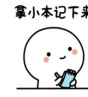

Figure 11: Some cases in our proposed TG-2M dataset.

million text lines. On average, each text line contains 4.83 characters. Notably, 97.8% of the text lines have fewer than 20 characters, which facilitates effective training of our model.

## B   Discussion about the Two-Stage Framework

We propose a two-stage generation framework that achieves detail optimization and task unification. However, this enhancement does not notably improve recognition accuracy in our ablation study. This is because: 1) The first stage already allows for some detailed modifications. 2) Our ablation study was conducted on a subset of the TG-2M dataset. The second stage enhances texture details, but its performance is limited with insufficient data. On the complete dataset, the two-stage framework demonstrates better performance, as detailed in Table 4.

Table 4: The comparison of performance on the complete dataset.

| Methods      | English |        | Chinese |        |
|--------------|---------|--------|---------|--------|
|              | ACC↑    | NED↑   | ACC↑    | NED↑   |
| w/o TwoStage | 71.11   | 88.07  | 66.68   | 83.16  |
| w TwoStage   | **73.36** | **88.98** | **67.92** | **83.94** |

## C   Discussion about FID

The Fréchet Inception Distance (FID) metric evaluates the distribution gap between generated images and target images. A higher FID indicates a larger distribution gap. We evaluate the FID score on the AnyWord [28] benchmark, which provides a subset of images for this purpose. Since the AnyWord FID benchmark is derived from the AnyWord training set, it is reasonable for AnyText to achieve a better FID score due to the distribution gap between our TG-2M and AnyWord. Additionally, our TextGen can generate more artistic texts, resulting in a diversity of distribution. Therefore, a lower FID score does not necessarily imply a lower visual quality of the generated images.

## D   Discussion about Future Work

Based on ControlNet, current visual text generation has made significant progress. We further investigate the control information in ControlNet-based visual text generation tasks and draw several conclusions. Future performance improvements can be achieved through three approaches: 1) Construct high-quality datasets, as current datasets still contain incorrect labels and unreasonable captions. 2) Enhance the text embedding module. Leveraging large language models (LLMs), we can

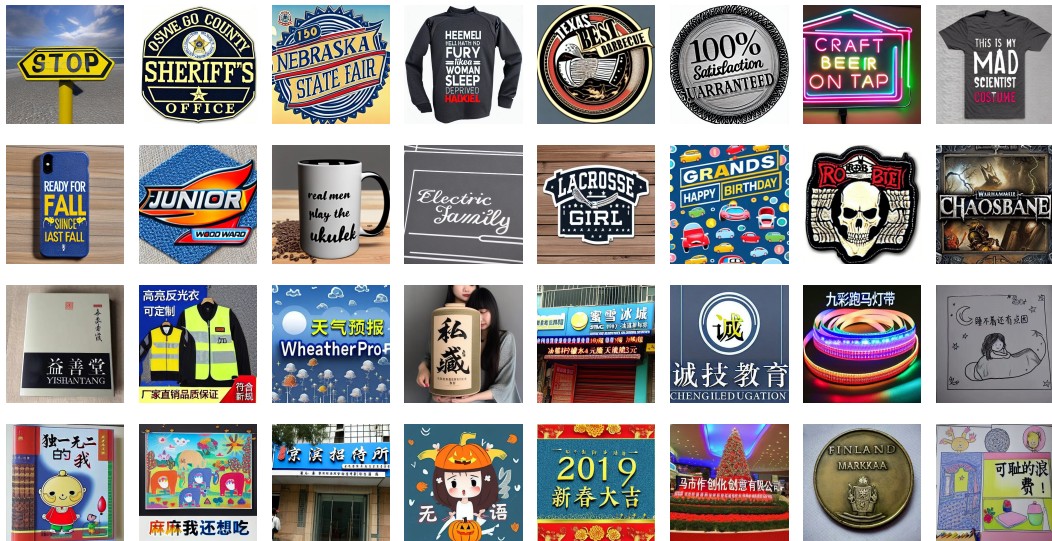

Figure 12: More qualitative results generated by our TextGen. Both Chinese and English are high-quality and realistic.

design a more robust text embedding module than the CLIP text encoder, capable of understanding more detailed captions.

# E More Qualitative Results

We present additional qualitative results generated by TextGen in Fig. 12. TextGen produces realistic images with coherent and readable text. Additionally, TextGen is capable of generating artistic text for applications such as logos, posters, and clothing design.

