# OpenReview forum: "How Control Information Influences Multilingual Text Image Generation and Editing?"
_NeurIPS.cc/2024/Conference — NeurIPS 2024 poster_

### Official Review · Reviewer_dBLt · 2024-06-20

**Soundness:** 3
**Presentation:** 3
**Contribution:** 3
**Rating:** 7
**Confidence:** 4

**Summary:**

This article investigates an intriguing issue: how controlling information affects performance in the process of text image editing and generation. Through a series of observations and experiments, the authors extract valuable insights. Building upon these findings, they craft a novel framework that demonstrates impressive performance in both the generation and editing of text images.

**Strengths:**

1.	The experimental design, performance analysis, and underlying motivations are presented in a systematic manner with high-quality writing. Some findings offer valuable guidance for the community.
2.	The performance gain in recognition accuracy is substantial.
3.The dataset introduced holds significant value.

**Weaknesses:**

1.	There is an absence of detailed quantitative analysis for the internal workings of the designed modules.
2.	Although some perspectives are effectively illustrated through visualizations, they lack the backing of quantitative analysis.
3.     Typos in references, such as Diffute: Universal text editing diffusion model. Trivial writing of the experimental section.

**Questions:**

1.	Within an equitable comparison framework, the performance improvement in TextGen on the FID metric is less pronounced compared to its advances in recognition accuracy. Are there any further analyses on this issue?
2.	The author notes, "Although TG2M... it is highly effective for training and achieves superior performance." Is there any quantitative evidence available to support this claim?

**Limitations:**

The authors list the potential limitations and analyze the reasons briefly.

---

> ### Author Rebuttal · Authors · 2024-08-06
>
> We thank you for your valuable comments and positive attitude to our paper. Below are my rebuttal and discussion for these questions 😊.
>
> **Question 1: Internal workings.** The pipeline contains two control models and a diffusion process. As shown in Figure 4, the early timestamps in diffusion process using global control model and the latter timestamps in diffusion process using local control model.
>
> **Question 2: Quantitative analysis for perspectives.** For the control input, the first line in Table 1 presents a general diffusion model using ControlNet, which performs worse than our proposed methods. This indicates that general diffusion cannot generate accurate text. More visualizations will be added in the Appendix in the final version of our paper. For control at different stages, we conducted an additional experiment by removing control at various stages. As shown in the table below, removing early control information results in a slight performance decrease, whereas removing late control information leads to a significant performance decline. This supports our conclusion that early control information is crucial for generating coherence and realism, while late control information ensures text detail and accuracy. For the control output, we calculated the frequency distribution shown in Figure 5 and performed an ablation study presented in Table 1 (lines 4 and 5).
>
> |                         |  English  |  Chinese  |
> |-------------------------|:---------:|:---------:|
> | Baseline                |   60.18   |   61.42   |
> | Removing early control  |   54.36   |   53.28   |
> | Removing late control   |   34.84   |   23.42   |
>
> **Question 3: Incorrect citation.** We apologize for the incorrect citation, which may have occurred due to an error during the copying from Google Scholar. We will correct this mistake in the final version of the paper.
>
> **Question 4: The FID score.** The FID score measures the feature similarity between generated images and target datasets. We evaluate the FID score using the AnyWord-FID benchmark. 1) A higher FID score does not necessarily indicate lower visual quality of the generated images. This conclusion has been noted in earlier works [1]. Our generated images exhibit greater diversity, such as word art, which is scarce in the AnyWord-FID dataset, resulting in a higher FID score, as described in [1]. 2) The images in the AnyWord-FID benchmark are selected from the AnyWord training set, while we use a different training set, making it reasonable for AnyText to achieve a better FID score.
>
> **Question 5: The evidence for dataset effectiveness.** Our TG2M dataset contains only 2.5M images, which is significantly smaller than other training sets. As shown in Table 2, our baseline model is a general diffusion model similar to AnyText [2], trained for only 5 epochs, in contrast to other methods trained for 10 epochs. It can be observed that using our TG2M dataset achieves performance comparable to other methods despite using fewer data and fewer epochs. Recently, we employed more stringent filtering methods to process and clean the data, resulting in higher-quality data. Such data can further enhance generation performance. We will fully release our entire dataset and code without reservation after the paper is accepted.
>
> [1] Li T, Chang H, Mishra S, et al. Mage: Masked generative encoder to unify representation learning and image synthesis[C]//Proceedings of the IEEE/CVF Conference on Computer Vision and Pattern Recognition. 2023: 2142-2152.
>
> [2] Tuo Y, Xiang W, He J Y, et al. Anytext: Multilingual visual text generation and editing[C]//Thirty-seventh Conference on Neural Information Processing Systems. 2023.

---

> > ### Comment · Reviewer_dBLt · 2024-08-07
> >
> > Most of my concerns have been addressed. One remaining issue is that the authors pointed out that FID can not indicate the visual quality of the generated images effectively. Is there any other quantitative metric about image quality that can verify the effectiveness of TexGen?

---

> > > ### Author Response · Authors · 2024-08-09
> > >
> > > Thank you for your positive response! **First**, I would like to clarify that the FID metric does assess image generation quality to some extent. However, the evaluation dataset used in my experiments is the FID subset of the publicly available AnyWord dataset, which was selected from the AnyWord training set. Consequently, AnyText naturally yields a lower FID score, so our comparisons are not really fair to us. As shown in Table 2 in our paper, our method achieves a better FID score compared to other models when AnyText is excluded. **Additionally**, since FID is widely considered an inadequate measure of image generation quality, as noted by many scholars, the question you have raised is a very valuable one and well worth thinking about. Personally, We think that aesthetic scoring offers a better evaluation method. Therefore, we employed Laion's publicly available aesthetic scoring model [1] to evaluate the images generated by our model, and we calculated the average aesthetic score of these images, as detailed in the table below.
> > >
> > > |                  | English | Chinese |
> > > |------------------|:-------:|:-------:|
> > > | AnyText  | 4.27   | 5.06   |
> > > |Ours | 4.42   | 5.18   |
> > >
> > > Lastly, we emphasize that your question is crucial and worth deep consideration by everyone. The solution we provided is one approach, and we hope other researchers will also explore improved methods for evaluating generation quality.
> > >
> > > [1] https://laion.ai/blog/laion-aesthetics/

---

> > > > ### Comment · Reviewer_dBLt · 2024-08-09
> > > >
> > > > Thanks for the response. Given the presented aesthetic scoring and previous replies, I will raise the score. (BTW, some of the most recent advancements in Blind Image Quality Assessment (BIQA) metrics can provide new perspectives in evaluating the quality of generated images.)

---

> > > > > ### Author Response · Authors · 2024-08-14
> > > > >
> > > > > We sincerely appreciate your positive feedback and strong support for our work. Thank you also for suggesting the BIQA metrics; we will thoroughly explore this perspective in the future.

---

### Official Review · Reviewer_Z5LF · 2024-07-12

**Soundness:** 3
**Presentation:** 4
**Contribution:** 4
**Rating:** 7
**Confidence:** 4

**Summary:**

This paper introduces a novel framework to enhance the quality of multilingual text image generation and editing by optimizing control information. The authors investigate the impact of control information from three perspectives: input encoding, the role at different stages, and output features. They find some insight conclusion and propose a framework (TextGen) employs Fourier analysis to emphasize relevant information and reduce noise, uses a two-stage generation framework to align the different roles of control information at various stages. Furthermore, they introduce an effective and lightweight dataset for training. The method achieves state-of-the-art performance in both Chinese and English visual text generation.

**Strengths:**

1. The paper investigate the influence of control information, which can greatly inspire future works of the community. In summary, this paper is well organized, with reasonable motivation and insights.
2. The use of Fourier analysis to enhance input and output features, along with the two-stage generation framework, offers innovative methodological approaches in the field of text image generation.
3. The creation of a new lightweight yet effective dataset, TG2M, provides a valuable resource for training in visual text generation and editing.

**Weaknesses:**

1. To my best knowledge, there exist other benchmarks such as the benchmark in TextDiffuser[1]. Although these benchmarks may not be of good quality, it would be better to compare TextGen with other works on these benchmarks or construct a high-quality benchmark based on TG-2M.
2. The data building process is not described in detail. Even though there are some differences (the recaption using Qwen-VL, the data selection), it looks a bit similar to AnyWord-3M and the author needs to describe the differences in paper or Appendix.

[1] Chen, Jingye, et al. "TextDiffuser: Diffusion Models as Text Painters" NeurlPS 2023
[2] Tuo, Yuxiang, et al. "AnyText: Multilingual Visual Text Generation And Editing" ICLR 2024

**Questions:**

1. Could TextGen generate other languages, such as Korean?
2. How about the diversity of textgen's generation of the same prompt? The authors can put some visualization examples.

**Limitations:**

refer to weaknesses.

---

> ### Author Rebuttal · Authors · 2024-08-06
>
> We thank the reviewers for their positive attitude towards our paper and for their valuable comments 😊. Below are our rebuttal and discussion of the questions.
>
> **Question 1: The evaluation.** We conducted evaluations on the AnyWord benchmark, which is the highest quality benchmark. Your suggestion is valuable. We recently employed more stringent filtering methods to process and clean the data, resulting in higher-quality datasets. Such data can significantly enhance generation performance, and we have constructed a new benchmark from it. We will fully release our entire dataset and code without reservation after the paper is accepted.
>
> **Question 2: The data building pipeline.** We recaptioned the data using both BLIP and Qwen-VL. Due to the input length restriction in the CLIP text encoder, the diffusion model requires conditions shorter than 77 tokens. Qwen-VL often generates longer captions and sometimes hallucinates. Therefore, we generated the initial captions using BLIP, identified failure cases, and then recaptioned those using Qwen-VL. Additionally, we filtered the data using aesthetic scores and a watermark detection model.
>
> **Question 3: More languages.** Our training set only contains English and Chinese cases, which enables our model to generate high-quality English and Chinese visual texts. Moreover, by training with the glyph conditions, the model has developed a capacity to adhere to the glyph conditions. For other languages such as Korean and Japanese, the model can still generate them effectively in some cases when the glyph condition is provided. But as for more difficult languages such as Arabic, it is hard for our model to generate. We will try to construct a more comprehensive multi-lingual model in the future.
>
> **Question 4: The diversity.** Similar to other diffusion-based models, our model can generate diverse results when given different random seeds. Due to rebuttal restrictions, we are unable to provide visual examples here. We will include visual results related to diversity in the Appendix of the final version of our paper.

---

> > ### Comment · Reviewer_Z5LF · 2024-08-13
> >
> > Thanks for the response. All my concerns have been properly solved. I thus give ACCEPT as my final score.

---

> > > ### Author Response · Authors · 2024-08-14
> > >
> > > Thank you for your appreciation of our work.

---

### Official Review · Reviewer_JA2v · 2024-07-13

**Soundness:** 2
**Presentation:** 2
**Contribution:** 1
**Rating:** 5
**Confidence:** 5

**Summary:**

This study explores the advancement of visual text generation using diffusion models within a ControlNet-based framework, focusing on the impact of control information. It examines input encoding, the role of control information during different stages of the denoising process, and the resulting output features. The authors propose TextGen, a method that aims to enhance generation quality by optimizing control information with Fourier analysis and a two-stage generation process.

**Strengths:**

1. Analyzed the impact of three control factors on text image generation.
2. Achieved comparable results in text image generation and editing.

**Weaknesses:**

1. The method lacks novelty, the writing of the paper is unclear, and the experimental results are insufficient. The experimental analysis does not provide new insight into the field.
2. The paper's analysis of control factors for text image generation is limited to three control conditions, which may not be the most effective. For example, character embeddings could be used as input, sentence prompts through an LLM could be used, or text coordinate encodings could be used as control conditions.
3. The necessity of the FEC module is not very clear, as it does not specifically target text image scenarios. Additionally, the computational efficiency of FFT and IFFT needs to be considered.
4. Why are global control and detail control split into two stages? Would it not be possible to consider both global and detail control simultaneously during the whole diffusion process?
5. The improvement of this method over existing methods is not obvious, especially in terms of visual and perceptual quality. Training on a small dataset also affects the method's generalizability. For example, can it still correctly generate text that appears infrequently or not at all in the proposed dataset?
6. The results lack demonstrations of “scene” text image generation; most images resemble text printing and have less realism compared to Anytext. This may be due to dataset bias, as shown in Figure 11. Thus, a small dataset is not a contribution of this paper and could be considered a drawback for large models.
7. The ablation experiments are insufficient, and the effectiveness of the modules is not convincingly demonstrated.

**Questions:**

1. Why only consider these three control methods.
2. What is the reason and necessity for introducing Fourier transform.
3. Why can't both global and detail control be considered simultaneously during the diffusion process? Wouldn't it be possible to use a learnable module to control the weights of these two controls. The two seperated control stages proposed in this paper is clearly not optimal.
4. It is recommended to add more ablation experiments, such as without the Fourier module, and considering both global and detail control simultaneously. More visualization results should be provided, such as visualizations of the ablation experiments.
5. How does the method perform on characters that appear infrequently in the dataset.
6. It is recommended to include failure cases and analyze the reasons. Additional results for text image editing should also be provided.
7. Please adequately address the concerns raised in the weaknesses section.

**Limitations:**

Yes

---

> ### Author Rebuttal · Authors · 2024-08-06
>
> We appreciate your insightful comments and have carefully considered your questions and suggestions. Below is our rebuttal and discussion of these questions 😊. Due to rebuttal restrictions, we are unable to include images. We will provide the corresponding visual results in the Appendix of the final version of our paper.
>
> **Question 1: Why only consider these three control methods.** Our research is based on currently common visual text generation networks such as AnyText [1] and TextDiffuser [2]. These methods typically require glyph images for guidance. Since existing methods do not consider some unique properties of text, we investigated the impact of this control information. Additionally, *we examined the influence of the same control information from three different aspects, rather than studying three control methods.*
>
> **Question 2: Fourier transform.** Our motivation for employing FEC is detailed in Sections 1, 3.1.1, and 3.3 of the paper. We analyze it more clearly below. **1)** As shown in Figure 2, general ControlNet conditions primarily affect macroscopic styles and edges and minor texture inaccuracies are acceptable, but *visual text generation requires precise control over textures to avoid content errors and unrealistic details.* **2)** Text glyph conditions are sparse, with most areas being black, acting as noise in standard spatial convolutions. *The Fourier transform filters frequency*, enhancing attention to specific components in detail-rich areas of glyph images. **3)** Direct convolution in the spatial domain is limited by the receptive field, while each point in the frequency domain captures global information of the same frequency. *Convolution in frequency domain enables global interactions among similar frequency components.*
>
> **Overall, while such an approach is unnecessary for general generation, it is crucial for text generation.** Experimental results in Table 1 further confirm the necessity of utilizing the Fourier transform.
>
> **Question 3: Simultaneously global and detail process.** **1)** Our objective is to *investigate the differences between early and late control stages*, as compared to general scene images. This guides us to employ some strategies to promote the model’s learning. **2)** We have implemented a method that combines both local and global control, using MLP to generate control factors that weight both components. However, challenges in balancing these control factors during gradient descent led to noticeable oscillations in training, ultimately resulting in degraded generation quality. **3)** Using both global and local modules would **double the training cost** and significantly decrease the model's speed, thereby limiting its generalization capability. Consequently, the two-stage method proposed in this paper emerges as the optimal solution.
>
> **Question 4: More ablation studies.** **1)** The ablation study without the Fourier block has been conducted in Table 1, lines 1 and 2. Results show that our FEC significantly improves generation quality, especially for Chinese text, due to its complex edges and details. **2)** The ablation study considering both global and detailed control simultaneously is shown below. This model requires a weighting mechanism to balance the two conditions, posing a challenge that we discussed in detail in our response to question 3.
>
> ||English|Chinese|
> |-|:-:|:-:|
> |Two stage model|60.18|61.42|
> |One stage model|58.64|59.56|
>
> **Question 5: Characters frequency.** By training with the glyph conditions, the model has developed a capacity to **adhere to the glyph conditions**. For characters that are infrequently represented in the dataset, the model can still generate them effectively when the glyph condition is provided. The model even demonstrates some generation capability for languages that are not present in the training set. We will add relevant visual results in the Appendix.
>
> To further support this, we calculated the character generation accuracy of our model for both high-frequency and low-frequency characters, as shown in the table below. The difference in accuracy is not significant.
>
> ||high-frequency characters|low-frequency characters|
> |-|:-:|:-:|
> | Accuracy|61.95|58.41|
>
> **Question 6: Failure cases.** Our model encounters failure cases in generating small text due to VAE limitations, which is not the focus of this paper (detailed in Section 5). Some text generation errors also stem from poor-quality training data. Despite our data being of the highest quality, it contains some incorrect detections and captions, limiting the model's capabilities. We have recently applied stringent filtering methods to improve data quality, which can significantly enhance performance. We will release the entire dataset and code after the paper is accepted.
>
> **Question 7: Dataset.** Many recent works, such as LLaVA [3], propose that the quality of training data is essential for a model, even if the quantity is small. We believe it is highly significant to construct high-quality datasets, even if they are relatively small. Additionally, our model demonstrates superior performance on small datasets. When the amount of data is increased, the model can achieve even greater performance. **However, scaling up is not our primary goal.**
>
> **Question 8: Scene generation cases.** We list more generated results in the Appendix. As shown in Figure 12, there are many scene cases. We can also generate realistic images.
>
> [1] Tuo Y, Xiang W, He J Y, et al. Anytext: Multilingual visual text generation and editing[C]//Thirty-seventh Conference on Neural Information Processing Systems. 2023.
>
> [2] Chen J, Huang Y, Lv T, et al. TextDiffuser: Diffusion Models as Text Painters[C]//Thirty-seventh Conference on Neural Information Processing Systems. 2023.
>
> [3] Liu H, Li C, Li Y, et al. Improved baselines with visual instruction tuning[C]//Proceedings of the IEEE/CVF Conference on Computer Vision and Pattern Recognition. 2024: 26296-26306.

---

> > ### Comment · Reviewer_JA2v · 2024-08-13
> >
> > Thank you for your response. The author addressed most of my concerns. I also read the questions and responses from other reviewers in detail. I have improved my score, but I still think that this article lacks sufficient novelty.

---

> > > ### Author Response · Authors · 2024-08-14
> > >
> > > We sincerely appreciate your timely feedback and the strong support for our work. We are committed to incorporating all of the clarifications you suggested in the next version of our paper.

---

### Official Review · Reviewer_bu9j · 2024-07-18

**Soundness:** 2
**Presentation:** 3
**Contribution:** 2
**Rating:** 3
**Confidence:** 4

**Summary:**

This paper analyzes the issue of control in image generation models. Specifically, the article addresses three aspects: input control information, the impact of control information at different stages, and output control information. The model was optimized for two tasks: text-to-image and image editing. Using a method similar to FreeU, the paper conducts Fourier frequency domain analysis on the input and output features and proposes a two-stage generation model based on previous findings.

**Strengths:**

1.The writing of the article is excellent, with clear and concise sentences and well-organized structure.
2.The proposed final model FEC+GP+TS+IFE shows significant improvement.
3.The method is quite innovative, with a novel approach.

**Weaknesses:**

1.Some models lack detailed theoretical analysis, which makes the purpose of the model unclear.
2. It is not clear what the task of this paper issue. Please point it out in a prominent position in the first chapter.
3. The layout is a bit messy. For example, should Table 2 be placed above Table 1?

**Questions:**

1.Will the dataset be released? If so, please address the privacy, copyright, and consent of the data.

2.The convolution theorem states that the Fourier transform of a function's convolution is the product of the Fourier transforms of the functions. In other words, convolution in one domain corresponds to multiplication in another domain, such as time-domain convolution corresponding to frequency-domain multiplication. Can the FEC network in the article be considered as performing a simple point-wise multiplication operation?

3. I have questions about the effectiveness of FEC. The article only shows that using FEC is better through ablation experiments, but I have two concerns: 1) The improvement for English is not significant, while it is more noticeable for Chinese. Is the main reason for this that the model was fine-tuned on Chinese? 2) When using methods like depth control, the output might not match the control, but the output could still be reasonable. Using glyph images for control requires a strong match between the output and the input text. From a human observation perspective, glyph control might seem more precise, but for model training with MSE loss, both approaches could be similar.

4. Section 4.3 does not have accompanying text?

5. Why does convolution perform poorly in understanding text? Shouldn't there be some small experiments to demonstrate this? Alternatively, could it be explained through some visualization analysis or derivation process?

6. Figure 4, which serves as the overall pipeline diagram, does not seem to reflect the inference structure discussed in Chapter 3.5.

**Limitations:**

The article does not specifically discuss the potential negative societal impacts. For example, generating high-quality images guided by text might facilitate the creation and spread of fake news. Additionally, using more complex models could lead to increased resource consumption, among other issues.

---

> ### Author Rebuttal · Authors · 2024-08-06
>
> We thank you for your valuable comments and some of the affirmations of our paper. These questions are insightful and deepen my thinking. Below are my rebuttal and discussion for these questions 😊.
>
> **The writing.** We apologize for the writing errors in our paper. In this paper, we primarily studied the impact of control information. Through experiments, we derived some conclusions regarding control information. Based on these conclusions, we designed a novel and effective model to address the issues. We will clear the presentation in the final version of the paper, specifically in the first chapter, according to your suggestions. Furthermore, we believe that the placement of Table 1 and Table 2 is appropriate, as the body of the paper first presents ablation experiments followed by comparative experiments. If you find this sequence unreasonable, we will adjust the order in the final version of the paper.
>
> **Question 1: Privacy and copyright.** We recently employed more stringent filtering methods to process and clean the data, resulting in higher-quality data. Such data can significantly enhance generation performance. We will **fully** release our all dataset and code **without reservation** after the paper is accepted.
>
> **Question 2: Convolution and Fourier transform.** In mathematical theory, convolution in the frequency domain is equivalent to multiplication in the spatial domain. However, in our model, performing convolution in the frequency domain exhibits some differences. This is discussed in Section 3.3 of our paper, but we offer a more detailed discussion here:
> 1) In the frequency domain, **additional operations such as activation functions are applied alongside convolutions**, making it not strictly equivalent to point-wise multiplication in the spatial domain.
> 2) Direct convolution in the spatial domain is limited by the receptive field. However, each point in the frequency domain represents global information of the same frequency. This allows convolution in the frequency domain to enable **global interactions of similar frequencies**.
> 3) Performing convolution in the frequency domain can capture features that are difficult to learn in the spatial domain during gradient descent training. **The training effect of using gradient descent is not equivalent to the point-wise product operation in the spatial domain.**
>
> **Question 3: Concerns in FEC.**
> 1. **Concern 1:** The data and data mixture used in the ablation study in Table 1 are entirely consistent with Anytext [1], which randomly selected 200k training samples from the dataset. We did not perform any separate fine-tuning. The noticeable improvement in Chinese data is due to the model's enhanced sensitivity to details under the frequency domain enhancement from FEC. For generating more complex Chinese content, the performance improvement achieved with less data will be even greater.
> 2. **Concern 2:** Using MSE loss, the model's training objective remains consistent. However, we propose to enhance the model's ability to perceive details and high frequency. This allows the model to leverage more detailed information, thereby improving the quality of detail generation under the same MSE loss. While such an approach is unnecessary for depth-controlled generation, it is crucial for text generation, which demands fine-grained control.
>
> **Question 4: Section 4.3.**  We apologize for the typographical errors and we will revise the typography in the final version of the paper.
>
> **Question 5: Discussion about convolution.** Our argument is not that convolution performs poorly in understanding text, but that standard convolution faces challenges. Due to the restricted receptive field of convolution, conventional operations struggle to capture global information. **Text often appears in elongated forms**, covering a large area in one direction, which hinders the performance of ordinary convolution in extracting text features and semantics. **Similar observations have been noted in several scene text recognition studies [2] [3]**. Moreover, in glyph images, **most regions are purely black**, serving as noise relative to the text regions during standard spatial convolution operations. We will reference the aforementioned studies to support our perspective in the final version of the paper.
>
> **Question 6: Question about the figure.** The figure 4 contains the inference pipeline we proposed. The lower part of the diagram illustrates the inference strategies. Different inference strategies are used for generation and editing tasks. The color of the UNet corresponds to either the global or detail ControlNets, indicating the use of different control models at various stages. The diagram is consistent with the description in Section 3.5.
>
> [1] Tuo Y, Xiang W, He J Y, et al. Anytext: Multilingual visual text generation and editing[C]//Thirty-seventh Conference on Neural Information Processing Systems. 2023.
>
> [2] Fang S, Xie H, Wang Y, et al. Read like humans: Autonomous, bidirectional and iterative language modeling for scene text recognition[C]//Proceedings of the IEEE/CVF conference on computer vision and pattern recognition. 2021: 7098-7107.
>
> [3] Atienza R. Vision transformer for fast and efficient scene text recognition[C]//International conference on document analysis and recognition. Cham: Springer International Publishing, 2021: 319-334.

---

> ### Author Response · Authors · 2024-08-14
> **Sincere Invitation to Participate in the Discussion**
>
> Dear Reviewer bu9j,
>
> We sincerely appreciate your time and feedback. Given the rush in finalizing the writing, some aspects may have caused confusion or misunderstanding. It is our priority to ensure that the rebuttal aligns with your suggestions, and we are open to further discussions to clarify any remaining questions or concerns. We would be grateful if you could improve the evaluation after reviewing our responses.
>
> Thank you very much for your consideration.
>
> Sincerely,
>
> The Authors

---

### Decision · Program_Chairs · 2024-09-25

**Decision:**

Accept (poster)

**Comment:**

This paper presents a dataset and approach for dual-lingual image-to-text generation and editing.  This paper was reviewed by three experts, who overall lean towards acceptance.  While some questions as to the novelty of the approach remain, it can at least be seen as an application that may be relevant to some researchers.  The authors are encouraged to take the reviewers comments into account when creating their final version.